# Finite Volume Informed Graph Neural Network for Myocardial Perfusion Simulation

**Raoul Sallé de Chou**[1,2]                                    RAOUL.SALLE-DE-CHOU@INRIA.FR
**Matthew Sinclair**[3]                                          MSINCLAIR@HEARTFLOW.COM
**Sabrina Lynch**[3]                                               SLYNCH@HEARTFLOW.COM
**Nan Xiao**[3]                                                      NXIAO@HEARTFLOW.COM
**Laurent Najman**[4]                                          LAURENT.NAJMAN@ESIEE.FR
**Irene E. Vignon-Clementel**[*1]                  IRENE.VIGNON-CLEMENTEL@INRIA.FR
**Hugues Talbot**[*2]                              HUGUES.TALBOT@CENTRALESUPELEC.FR

[1] *Inria, Palaiseau, France*
[2] *CentraleSupelec, Inria, Université Paris-Saclay, France*
[3] *HeartFlow Inc., Mountain View, USA*
[4] *Univ Gustave Eiffel, CNRS, LIGM, France*

**Editors:** Accepted for publication at MIDL 2024

## Abstract

Medical imaging and numerical simulation of partial differential equations (PDEs) representing biophysical processes, have been combined in the past few decades to provide non-invasive diagnostic and treatment prediction tools for various diseases. Most approaches involve solving computationally expensive PDEs, which can hinder their effective deployment in clinical settings. To overcome this limitation, deep learning has emerged as a promising method to accelerate numerical solvers. One challenge persists however in the generalization abilities of these models, given the wide variety of patient morphologies. This study addresses this challenge by introducing a physics-informed graph neural network designed to solve Darcy equations for the simulation of myocardial perfusion. Leveraging a finite volume discretization of the equations as a "physics-informed" loss, our model was successfully trained and tested on a 3D synthetic dataset, namely meshes representing simplified myocardium shapes. Subsequent evaluation on genuine myocardium meshes, extracted from patients Computed Tomography images, demonstrated promising results, and generalized capabilities. Such a fast solver, within a differentiable learning framework, will enable to tackle inverse problems based on $H_2O$-PET perfusion imaging data.

**Keywords:** Graph Neural Network, Partial Differential Equations, Physics-informed Neural Network, Finite Volume method, Digital Twins, Perfusion simulation

## 1. Introduction

In (Li et al., 2023), a GNN with a finite volume constrained loss combined with ground truth data, successfully predicted the 2D Navier-Stokes solutions over a diverse range of shapes, starting from the solution of one time step and predicting the next one. (Donon et al., 2020) developed a FEM-informed loss function that effectively addressed the Poisson equation on various 2D shapes, employing a GNN without reliance on ground truth data. These neural

---

* Contributed equally

networks demonstrated in some cases superior generalization abilities compared to data-driven-only models, but were only tested in 2D cases, with a small number of unknowns and with simple shapes.

Our study introduces a Finite-Volume Informed Graph Neural Network for solving the 3D Darcy equations to model myocardial perfusion. It is trained on a synthetic dataset with diverse non-convex shapes resembling patient left ventricular geometries. Our model incorporates a physics-informed loss, thus eliminating the need for ground truth data. The PDE is discretized with Finite Volumes. Once the training is complete, the model can be applied in inference mode to new geometries to predict a solution, without retraining. We show preliminary results of our model and its generalization ability on a patient-specific left-ventricle myocardium dataset from coronary computed tomography angiography (CCTA) images. To our knowledge, this is the first physics-informed model tested on a real-world 3D dataset with a clinical purpose, and the first fully-based finite volume-informed neural network (FVINN), hence without the need for ground-truth data.

## 2. Datasets generation

### 2.1. Myocardial perfusion simulation

In prior investigations, (Papamanolis et al., 2021) formulated a patient-specific simulation pipeline for diagnosing coronary artery diseases based on CT-scans. The model is initiated with the segmentation of the aorta, epicardial coronary arteries, and left ventricle myocardium from CCTA image data. To account for small arteries that fill the myocardium volume and are too small to segment on CCTA images, synthetic coronary trees are generated from the epicardial segmented roots (Jaquet et al., 2018). Reduced order blood flow equations are then solved in the defined vascular network, with both segmented and synthetic components, to simulate the blood flow supply to the left ventricle myocardium.

Subsequently, to simulate perfusion in the myocardium and to account for the microcirculation (blood flow in the arterioles, capillaries, ...), a porous-media model governed by Darcy's law is utilized (Chapelle et al., 2010). The model comprises a source term driven by coronary flow and a homogeneous sink term (to simulate an ideal venous system). In order to obtain the source term, $p_{\text{source}}$, maps are generated by associating a perfusion territory in the myocardium with each outlet of the synthetic coronary tree. These territories are estimated through a Voronoi tessellation. Finally, a $p_{\text{source}}$ value is assigned to each perfusion region. This value is the pressure at the vasculature outlet, determined by solving the flow equations in the network.

We will focus in this study on predicting the FVM solution of the Darcy equations on different meshes and $p_{\text{source}}$ maps, by a FVINN. We describe in the following subsection the different datasets for training, validation, and test.

### 2.2. Synthetic dataset

To facilitate model training, we generated a synthetic dataset comprising half-ellipsoidal shells aimed at simulating simplified myocardium morphologies. The dataset was created using gmsh (Geuzaine and Remacle, 2009). As illustrated in Figure 1, the geometries were generated by defining 5 points connected by elliptical arcs. The resulting 2D shape was then

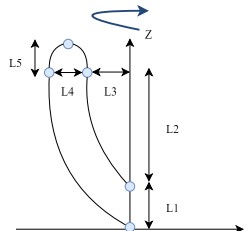 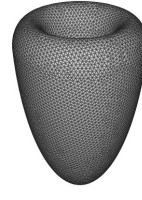 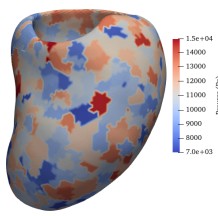

Figure 1: Left: schematic of the synthetic shape generation. Centre: example of one final synthetic 3D mesh. Right: example of one patient myocardium with its $p_{\text{source}}$ map.

extruded around the z-axis to form a volume, within which a volume mesh was generated. For the training dataset, 2000 distinct geometries were generated, with an additional 400 for the validation dataset. Various lengths $L1, L2, L3, L4$, and $L5$ were sampled to diversify the geometries. Furthermore, different $p_{\text{source}}$ maps were assigned to each geometry: within each volume, 1000 seed points were sampled. A Voronoi tessellation was grown from these points to emulate the perfusion regions associated with network outlets. Each Voronoi region was assigned a random $p_{\text{source}}$ value sampled from a normal distribution, in line with the observed $p_{\text{source}}$ distribution from patient-specific simulations (Papamanolis et al., 2021).

### 2.3. Patient myocardium data

As a test dataset, the left ventricle myocardium of 400 patients were segmented from CCTA image data, as part of the human-in-the-loop process for the HeartFlow $\text{FFR}_{\text{CT}}$ Analysis (HeartFlow Inc., US). CCTA images were received from multiple sites around the world by HeartFlow, Inc. (Taylor et al., 2013). 3D tetrahedral meshes of the myocardium volumes were generated with gmsh. Synthetic $p_{\text{source}}$ maps were also attributed to each myocardium mesh according to the previously explained method 2.2. Each mesh was composed of approximately 60k nodes in both datasets.

## 3. Methodology

### 3.1. Governing equations

In this work we focus on solving the 3-dimensional (3D) single compartment Darcy model for an incompressible flow:

$$\mathbf{w} + \mathbf{K}\nabla p = 0 \text{ in } \Omega \tag{1}$$

$$\nabla.\mathbf{w} = \beta_{\text{source}}(p_{\text{source}} - p) - \beta_{\text{sink}}(p - p_{\text{sink}}) \text{ in } \Omega \tag{2}$$

$$\frac{\partial \mathbf{w}}{\partial n} = 0 \text{ in } \partial\Omega \tag{3}$$

with $\mathbf{K}$ the permeability tensor, $\mathbf{w}$ the Darcy velocity, $p$ the capillary bed pressure, $p_{\text{source}}$ and $p_{\text{sink}}$ the source and sink pressure terms respectively, $\beta_{\text{source}}$ and $\beta_{\text{sink}}$ parameters describing the conductance of flow entering and exiting the myocardium respectively. These two parameters are patient specific; their computation is described in (Papamanolis et al.,

2021). By applying the divergence operator to Equation (1), we obtain the following Poisson equation, which can be solved for $p$ with no-flux boundary conditions (Equation (3)):

$$K\Delta p = -\beta_{\text{source}}(p_{\text{source}} - p) + \beta_{\text{sink}}(p - p_{\text{sink}}) \text{ in } \Omega \tag{4}$$

## 3.2. Finite volume method

FVM is a discretization method for the approximation of PDE. FVM is based on a volume integral formulation of the PDE over a finite number of control volumes discretizing the domain where the equations are solved in. The methodology unfolds by integrating the equations over each control volume. Introducing the vector $n$, normal to the control volume surface, and applying the divergence theorem on the diffusion term allows the replacement of volume integrals with surface integrals:

$$\int_S K\nabla p.ndS = \int_\Omega -\beta_{\text{source}}(p_{\text{source}} - p) + \beta_{\text{sink}}(p - p_{\text{sink}}) \, d\Omega \tag{5}$$

In this study, we consider the vertex-centered FVM where the control volumes are delimited by the dual mesh of a 3D tetrahedron element mesh (see Figure 2). Considering the mean value approach, the flux values (left-hand side of the equation here) are approximated by their values on the centre of each surface while the source (right-hand side) are approximated by their value at the centre of the control volume ($p$ in the following is the discrete pressure):

$$K \sum_{f \in \text{faces}} \nabla p_f.n_f S_f = (-\beta_{\text{source}}(p_{\text{source}} - p) + \beta_{\text{sink}}(p - p_{\text{sink}})) \, \Omega \tag{6}$$

The values on each face are linearly interpolated between the nodes connected to the edge crossing the face $f$. First order approximations are used for the pressure gradient.

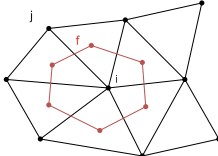

Figure 2: Example of 2D vertex-based control volume where the element centroids are connected to the edges centroids. In this study, generalization of such control volume to 3D tetrahedral meshes by replacing edges centroids by face centroids is employed. i is the indice of the node related to the control volume in red. f denotes one face of the control volume and j the indice of one neighbors of i.

## 3.3. Graph Unet Attention Network

### 3.3.1. GRAPH NEURAL NETWORK FOR SOLVING PDEs

GNNs represent a specific class of neural networks designed for processing graph structures. Specifically, meshes utilized for solving discretized PDEs can be viewed as graphs denoted

as $G = (N, V, E)$, where $N$ is the number of nodes, $V$ is a matrix of node features, and $E$ is the adjacency matrix indicating connections between nodes. GNNs typically consist of multiple layers that update the node feature matrix by exchanging information between nodes along graph edges.

The solution to the Darcy equations introduced in Section 3.1 necessitates solving a linear system of equations. Iterative methods, such as the Jacobi method, are very efficient to solve these large systems of equations. The Jacobi method starts with an initial guess of the solution and iteratively updates it based on the previous state:

$$p_i^{k+1} = \frac{1}{a_{ii}}(b_i - \sum_{j \in N_i} a_{ij} p_j^k) \tag{7}$$

With $p_i^k$ the Darcy pressure of node $i$ at iteration $k$, $N_i$ the indices of nodes belonging to node $i$ neighbourhood, $b_i$ and $a_{ij}$ coefficients computed from the different constants in 6. The solution is obtained once the difference between two successive states is small enough. As mentioned by (Liu et al., 2021), this sequence of operations is similar to the way GNNs perform.

### 3.3.2. GRAPH ATTENTION LAYER

The graph attention network was first introduced by (Velickovic et al., 2017). Given a graph $G = (N, V, E)$, for every node $i \in N$, attention coefficients are computed only between neighbours: $\alpha_{ij} = \text{softmax}_j(\text{LeakyReLU}(a^t[Wh_i || Wh_j])$. With $a$ a single-layer feedforward neural network, $W$ a weight matrix and $h_i$ the vector of node features of nodes $i$. The attention mechanism finally updates the feature vector $h_i$, similarly to the Jacobi method 7, through a nonlinear function $\sigma$:

$$h_i = \sigma(\sum_{j \in N_i} \alpha_{ij} W h_j) \tag{8}$$

### 3.3.3. GRAPH POOLING AND UNPOOLING

Similarly to (Zhao and Zhang, 2021), we employ a Graph Attention U-Net (GATU-net, Figure 3), with a slight deviation in the pooling and unpooling operations. The GATU-net architecture comprises three levels. During each pooling operation, a random sub-sample of the preceding upper-level mesh nodes serves as input. The node features of these sub-samples are computed by averaging the node features of the k-nearest neighbours from the upper mesh's level. Notably, the unpooling function is the one proposed in PointNet++ (Qi et al., 2017), where features are computed through distance-weighted interpolation from the k-nearest neighbours of the previous lower-level stage's mesh during the decoding phase.

A node encoder is employed to encode node features into a latent space. For each node, the encoder takes as input the node coordinates, along with values for $p_{\text{source}}$, $\beta_{\text{source}}$, and $\beta_{\text{sink}}$. Then, a node decoder processes the output of the GATU-net and predicts a pressure value for every node.

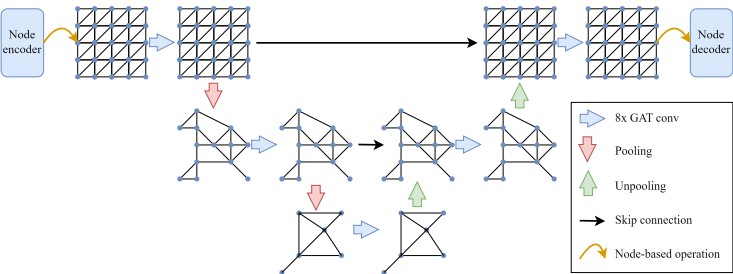

Figure 3: GATU-net architecture. Each grid corresponds to a mesh with a different node feature matrix computed thanks to the graph attention layers.

### 3.4. Finite volume informed loss function

The loss function to train the model is derived from Equation (7); it is the absolute mean over all nodes of the residual:

$$|p_\theta(i) - (\frac{1}{a_{ii}}(b_i - \sum_{j \in N_i} a_{ij}p_\theta(j)))| \tag{9}$$

With $p_\theta(i)$ the pressure output of the node $i$ of the decoder. It should be noted that no reference pressure solution is used to train the model. In addition, the same GATU-net architecture is trained with a classical L1 loss function between the network output and a ground truth from FVM. The latter model will be denoted as the supervised model (SM) in the next sections. The FVINN can be trained on a dataset comprising varying geometries and $p_{\text{source}}$ maps. The goal of the proposed method is to find a set of convolutions which minimize Equation (9) for every node of the meshes. Once trained, the model can be applied to a new geometry and $p_{\text{source}}$ map to predict the solution in inference mode.

### 4. Results

In this section, we present the results of FVINN and SM compared to the FVM solution. For both models, we report the actual, relative and normalized mean absolute error (MAE) and Mean max absolute error per shape (MMAE). The normalized error is the actual error divided by the standard deviation of the FVM solution. The Computational time to solve the linear system of equations given in Equation (6) with using the GPU-enabled pytorch function linalg.solve is approximately 4.28 seconds. The forward pass of our network to compute the solution takes 0.56 second.

The model was successfully trained on the synthetic dataset. SM obtained a slightly better MAE than FVINN on both the training and validation datasets. Specifically, on the validation dataset, the SM achieved a MAE of 3.74 Pa compared to 4.04 Pa for FVINN (Table 1). The relative MAE on the validation dataset was approximately 0.20% for both models, which corresponds to about 0.15 standard deviations of the FVM solution, as demonstrated by the normalized MMAE.

On the test dataset, the MAE was also smaller for the SM compared to FVINN, with normalized MAE values of 0.40 and 0.33, respectively. However, FVINN exhibited better

MMAE on all three datasets. Notably, the MMAE on the test dataset was 99.23 Pa and 35.10 Pa for SM and FVINN, respectively (Table 2). The high MMAE, combined with a relatively low MAE, indicates localized high errors for the SM, as displayed in Figure 4. No particular locations of these errors were found.

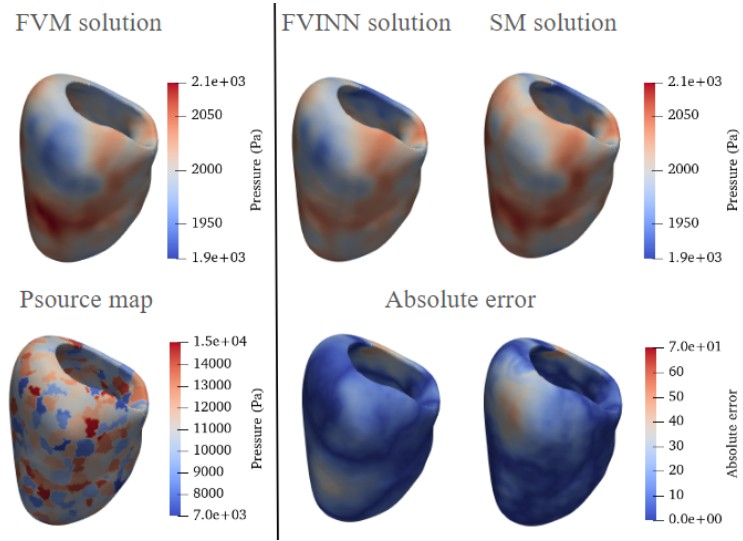

Figure 4: Comparison of the FVINN and SM results with the FVM ground truth on one case from the test dataset.

Table 1: Mean absolute error (MAE) and standard deviation of the absolute error on training, validation and test datasets.

| Dataset | Model | MAE (Pa) | Relative MAE (%) | Normalised MAE |
|---------|-------|----------|------------------|----------------|
| Training | FVINN | $4.02 \pm 1.19$ | $0.20 \pm 0.05$ | $0.16 \pm 0.04$ |
| | SM | $\mathbf{3.75} \pm 0.89$ | $\mathbf{0.18} \pm 0.04$ | $\mathbf{0.15} \pm 0.03$ |
| Validation | FVINN | $4.04 \pm 1.15$ | $0.20 \pm 0.05$ | $0.16 \pm 0.03$ |
| | SM | $\mathbf{3.74} \pm 0.86$ | $\mathbf{0.18} \pm 0.04$ | $\mathbf{0.15} \pm 0.07$ |
| Myo | FVINN | $8.06 \pm 2.28$ | $0.40 \pm 0.11$ | $0.31 \pm 0.08$ |
| | SM | $\mathbf{6.79} \pm 4.44$ | $\mathbf{0.33} \pm 0.18$ | $\mathbf{0.26} \pm 0.17$ |

Table 2: Mean max absolute error (MMAE) per shape and its stadard deviation on training, validation and test datasets.

| Dataset | Model | MMAE (Pa) | Relative MMAE (%) | Normalised MMAE |
|---------|-------|-----------|-------------------|-----------------|
| Training | FVINN | $\mathbf{18.50} \pm 6.94$ | $\mathbf{0.92} \pm 0.35$ | $\mathbf{0.75} \pm 0.20$ |
| | SM | $18.56 \pm 5.04$ | $0.93 \pm 0.30$ | $0.76 \pm 0.19$ |
| Validation | FVINN | $\mathbf{18.19} \pm 5.36$ | $\mathbf{0.90} \pm 0.26$ | $\mathbf{0.76} \pm 0.18$ |
| | SM | $18.31 \pm 1.68$ | $0.92 \pm 0.22$ | $0.78 \pm 0.21$ |
| Myo | FVINN | $\mathbf{35.10} \pm 11.70$ | $\mathbf{1.75} \pm 0.58$ | $\mathbf{1.37} \pm 0.41$ |
| | SM | $99.23 \pm 381.79$ | $3.92 \pm 5.75$ | $3.90 \pm 10.88$ |

## 5. Discussion

In this study, we presented a graph neural network for 3D myocardial perfusion prediction. The model incorporates a finite volume-informed loss function and requires no ground truth data for training. Successfully trained on a fully synthetic dataset, the neural network was then tested on patient-specific myocardium meshes. To our knowledge, this is the first application of such a model on a dataset of patient-specific data and large size meshes. Despite our model not being trained against ground truth from simulations, it achieves comparable accuracies on synthetic datasets, validating this approach. However, while SM obtained better MAE, the MMAE of our FVINN was better on the three datasets and almost three times smaller than SM on the patient-specific dataset. Our model demonstrates more robust generalization results on unseen geometries, which may differ significantly from the original distribution of shapes in the training dataset. Nevertheless, the synthetic datasets used in our study represent highly simplified myocardium shapes, which limits our ability to draw conclusions about the efficacy of FVINN or SM for predicting Darcy solutions on patient left-ventricle myocardium meshes. Future work should involve training these models on patient-specific or more complex synthetic data to further investigate their performance. In conclusion, our study shows promising results for developing deep learning-based models as PDE solvers for real-world applications. Beyond reducing computational complexity, models such as the FVINN offer the potential to solve inverse problems from $H_2O$-PET perfusion imaging data. Moreover, the integration of physics-based loss functions with an unsupervised framework holds potential for predicting more robust results while reducing the reliance on data, thereby supporting their application in medical context.

## Acknowledgments

Research funding for this project was provided by an industrial grant from Heartflow, Inc.

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

## Appendix A. Dataset Generation

The L1-L5 ranges were found empirically to represent the patient-specific LV volume ranges. The ranges in millimetres were [10, 20], [55, 85], [10, 20], [10, 20], [10, 20] for L1, L2, L3, L4 and L5, respectively. Uniform distributions were used within each range to sample new geometries. All the meshes were centred and align along z-axis.

## Appendix B. Supplementary materials for model architecture and training

The node encoder and decoder were simple feed-forward neural networks consisting of two hidden layers with 64 neurons each. LeakyReLU activation functions, along with batch normalization layers, were applied after every GAT convolution and layer of the node encoder and decoder.

For every pooling block of the GATU-net, the coarse mesh connectivity was computed as follows: clusters of nodes were generated by assigning each node of the upper level to

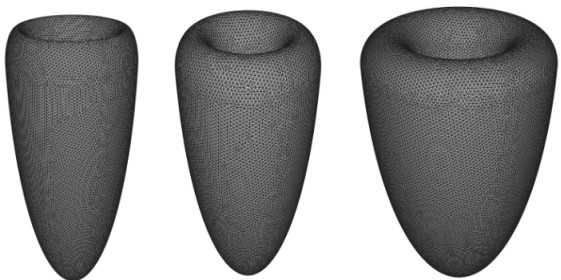

Figure 5: Example of meshes of the synthetic dataset.

the closest node of the sub-sample. Then, edges were established between two sub-sample nodes if their clusters were connected by an edge in the upper level.

We employed the Adam optimizer with an initial learning rate of 0.001 and a 0.99 exponential decay. Batches of 8 meshes were fed to the network during training for 400 epochs. The training lasted two days using two GPUs.

## Appendix C. Supplementary visual results

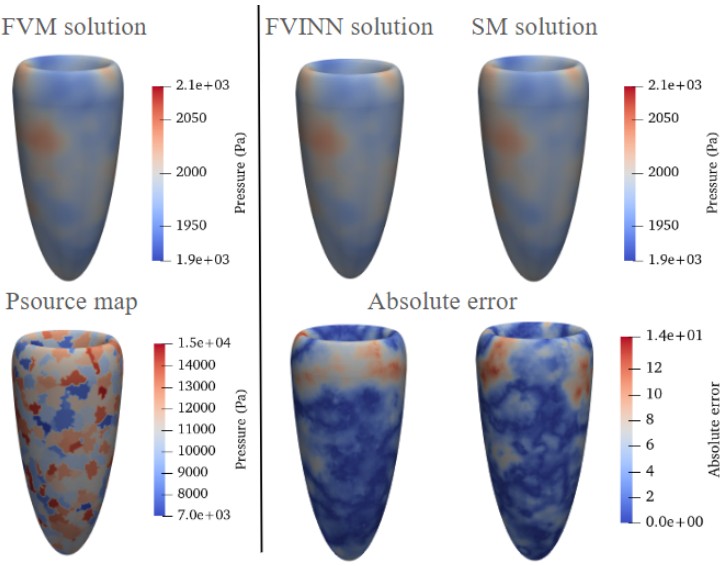

Figure 6: Comparison of the FVINN and SM results with the FVM ground truth on one case from the validation dataset.

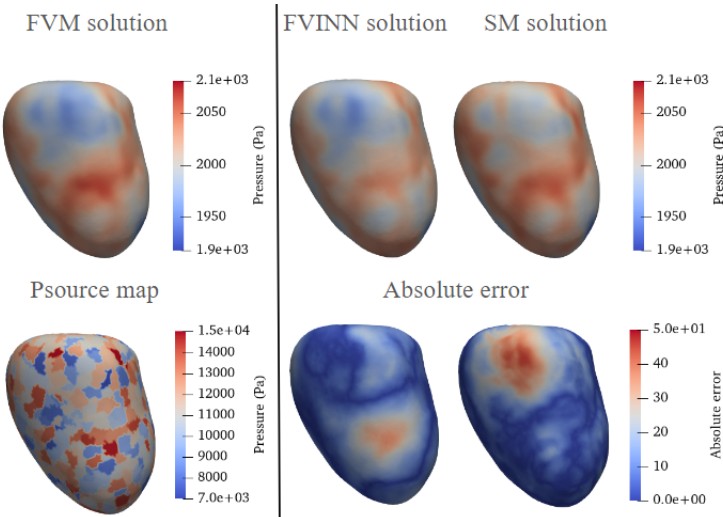

Figure 7: Comparison of the FVINN and SM results with the FVM ground truth on one case from the test dataset.

