# OpenReview forum: "Finite Volume Informed Graph Neural Network for Myocardial Perfusion Simulation"
_MIDL.io/2024/Conference — MIDL 2024 Oral_

### Official Review · Reviewer_4GD6 · 2024-02-27

**Confidence:** 4
**Preliminary Rating:** 2
**Recommendation:** Poster
**Final Rating:** 4

**Summary:**

The authors propose a novel method to accelerate the inference of myocardial perfusion simulations. Specifically, the method relies on a graph attention UNet, which is trained in an unsupervised manner by optimizing a physics-informed loss function derived from a finite-volume formulation. The method was trained on 2,000 synthetic  perfusion simulations with idealistic ventricular shapes and evaluated on additional simulations over 400 synthetic and 400 patient-specific geometries. The method achieves on both datasets a mean relative absolute error of 0.5% or lower.

**Strengths:**

- The proposed method is very interesting and another great example of physics-informed learning
- The method is novel in that it is the first fully unsupervised 3D finite-volume-informed graph neural network to be evaluated on both synthetic and patient-specific geometries
- The derivation of the finite-volume-based loss is well explained and may be understood even by non-experts
- The generalization to the patient-specific geometries is encouraging despite training the model on synthetic geometries (truncated ellipsoids)

**Weaknesses:**

The proposed method has several benefits, but I found several questions and concerns:

- First of all, the manuscript lacks essential information to replicate the results. Among other things, the paper is missing information on how the parameters for the synthetic geometries were sampled (e.g., range of parameters L1-L5 and the sampling pattern) as well as a clear description of the node encoder and decoder architecture is missing. Similarly, no hyperparameters for training the network were presented.

- Second, it seems to me like an essential limitation/requirement of the proposed method was not discussed at all. In particular, since the method takes as input nodal coordinates, the method would require the geometries to be in a canonical shape or it would require extensive augmentation to cover all possible rotations & translations.

- Similarly, Figure 1 suggests that relatively homogeneous meshes were generated for these experiments. It is therefore unclear what impact mesh resolution has onto the prediction results. Could the authors please also comment on how the receptive field size of the network relate to the overall size of the underlying mesh?

- One particular claim of the proposed method is to have improved generalization capabilities thanks to the physics-informed/finite-volume-informed graph neural network. There is, however, no comparator. While I understand that a fair comparison is not straightforward, I strongly believe that it would be very valuable if the method is at least compared against a regular physics-informed neural network and/or a supervised graph deep learning approach.

- In addition, the presented numerical evaluations only provide a global insight into the prediction accuracy. Neither the table nor the visual examples (since they are only showing a specific front of the geometry) can provide conclusions about any regional biases. Considering that cardiac ventricular geometries are considered, one simple next step may comprise the regional error analysis, e.g., based on the standardized 17-segment subdivision of the left ventricle by the American Heart Association. Moreover, it may be worth showing few more examples (front and back) of other heart geometries

- Finally, one particular advantage of such a method, as mentioned by the authors, may be the inference time once the network is trained. It would strengthen the paper, if a preliminary analysis on run-time differences between the ground truth finite-volume simulations and the network predictions (including the training time) were presented. In addition, I would appreciate if any information on the utilized software packages were presented.

**Detailed Comments:**

- All Equations: Shouldn't the Darcy velocity in 3D be a vector? If so, please properly reflect it in the equations
- Equation 4: I think there is a minus missing. Could the authors please double-check the equation?
- Figure 2: Please add a legend to clarify what f, i, and j are
- Section 3.3.2: The graph definition is different from the one in the preceding section, please revise the definition.
- Table 1: Two spaces are missing in the middle column. Please also clarify whether the values after +- denote the standard deviation or something else
- Figure 5: Please increase the size of the color bars and please use the same value range to color the meshes by the errors
- Abbreviations: Please be consistent with the abbreviations, use all capital letters for the abbreviations, and also define the abbreviations in the table descriptions so that the tables can be understood without having to read the entire paper
- Conclusion: non-supervised -> unsupervised

**Justification Of Final Rating:**

I appreciate that the authors have addressed all of my questions and concerns. The changes made to the manuscript have greatly improved the quality of the paper. The authors have now added all necessary information to replicate the results. I am therefore increasing the rating.

**Justification Of The Preliminary Rating:**

In general, this is a very interesting paper with a novel approach. In the current form of the manuscript, there is, however, too much essential information missing, which raises several question and concerns.

**Questions To Address In The Rebuttal:**

In addition to the questions and concerns raised above, I would appreciate if the authors could comment on the following topics:

- The manuscript states in the conclusion "Moreover, the integration of physics-based loss functions with a non-supervised framework holds potential for enhancing the explainability of the models while reducing the reliance on data, thereby supporting their application in medical context". From my point of view, neither the unsupervised learning nor the physics-based loss helps in any way with explainability since there will be no guarantee that the output of the network is physically plausible. It may, however, help with reducing the errors, improved generalization, and faster convergence. Could the authors please share their opinion on this topic?

- Equation 3 presents the boundary condition of the Darcy model. Were any special considerations in the loss function taken to properly handle/incoporate this constraint?

- In the description of the graph pooling, it was mentioned that a random subset of nodes is chosen. Could the authors please clarify the reasoning behind it and how the connectivity for the coarse graph was computed?

---

> ### Author Response · Authors · 2024-03-18
>
> We thank the reviewer for her/his insightful comments and positive feedback. All the comments are addressed below in the same order as they appear in the review:
>
> - The L1-L5 ranges were found empirically to represent the variability in patient LV volume ranges. The ranges in millimeters were [10, 20], [55, 85], [10, 20], [10, 20], [10, 20] for L1, L2, L3, L4 and L5, respectively. Uniform distribution within each range was used. The node encoder/decoder were simple feed-forward neural networks composed of two hidden layers with 64 neurons. Adam optimizer with initial learning rate of 0.001 and a 0.99 exponential decay was implemented. Batch of 8 meshes were passed to the network during training for 400 epochs. This information was added to the appendix.
> - The meshes were all centered and aligned along the z-axis using, for each geometry, the axis passing by the apex and the shape barycenter. Even though this data normalization method is not perfect, it reduces the need for data augmentation to cover possible rotations & translations.
> - We agree that the study of the impact of the mesh resolution should be conducted in further work. Experiments showed that approximately 30 graph convolutional blocks were necessary for the network receptive field to encompass the entire mesh. However, we found that, instead of using a single Graph Attention Neural Network with 30 convolutional blocks, a Graph U-net-like multiresolution architecture with 16 convolutional blocks on each level was performing even better. The pooling operations aim at increasing the receptive field.
> - We do not claim in the paper that we improve the generalization capabilities. We only talk about promising generalization capabilities: thanks to the FV-informed loss, the model shows good generalization on unseen geometry and p_source maps. This is in contrast to classical PINNs. We interpret those results as an improvement on the generalization capabilities of PINN. We realized an error on how we apply our activation functions in our model. Fixing the problem improved FVINN accuracy. The new results were added in the revised version of the paper. In addition, we provide a comparison of FVINN with a supervised model with the same network architecture implemented. While the supervised model obtained better MAE, it predicted higher MMAE overall on the all datasets. This last point indicates better robustness capabilities of our proposed FVINN compared to supervised models.
> - We agree that further work should comprise a regional error analysis based on the  standardized 17-segment subdivision of the left ventricle by the American Heart Association. Actually during the results analysis, many visual examples were analyzed and no regional bias was found. Additional visual results will be added in the appendix in the final version, if accepted.
> - It is well established that DL methods are faster at inference than classical numerical methods. To illustrate, we solve the linear system of equations in (7) using the GPU-enabled pytorch function linalg.solve the computational time is approximately 4.28s. The forward pass of our network to compute the solution takes only 0.56s. The model took approximately 2 days to train on two GPUs. An in-house code was used as the FVM solver. We have added this information in the revised version of the paper.
>
> - Indeed, the Darcy velocity is a 3D vector. Modifications was done accordingly.
> - Indeed, a minus was missing in equation (4).
> - Changes so that both definitions are the same were made.
> - The +- indeed denotes the standard deviation. Modifications were done accordingly.
> - Modifications were done accordingly.
> - Modifications were done accordingly.
>
> - Thanks to the FVM-derived loss function, we can constrain more what is learning the network and therefore increase the potential for the model to predict physically plausible solutions.
> - The boundary conditions mentioned in the Darcy model are incorporated in the model as source term on the mesh boundary element. As it is a zero-flux boundary condition, the source term to add is equal to zero. Therefore, no specific consideration in the loss function is required.
> - A random subset of nodes from the less coarse levels were chosen for memory and computational efficiency. Similarly to classical numerical methods, we could have generated coarser meshes for each level of the GATU-net. However, this would induce more computational cost to generate multiple meshes. This method would also require to store for every data, all the mesh nodes coordinates, therefore increasing the GPU memory usage and decreasing the training batch size which would affect the performance of the network. The coarse mesh connectivity is computed as follows: clusters of nodes are generated by assigning every node of the upper-level to the closest node of the sub-sample. Then, edges are built between two sub-sample nodes if their clusters were connected by an edge in the upper-level.

---

### Official Review · Reviewer_tawE · 2024-02-28

**Confidence:** 3
**Preliminary Rating:** 4
**Recommendation:** Oral

**Summary:**

The authors propose to train a physics-informed graph neural network (GNN) to solve the equations governing myocardial perfusion. The specific promise is that the model learns the physics purely from simulated data, yet generalises to real data sufficiently well. They show that this is qualitatively the case (the perfusion patterns match the ground truth obtained by "real" simulation), but quantitatively off by a factor.

**Strengths:**

I was impressed by the very detailed, accessible and well-structured and referenced introduction. I'm no expert in the field and can't properly assess if all relevant prior works have been cited, but the presentation is conclusive and matches my picture of the state of the art as much as I observed it.
The motivation of the contribution is likewise clearly presented and is in concordance with my own standpoint and experience, so I can follow this argumentation very well. I acknowledge from own discussions, though, that "pure mathematicians" don't easily jump on this mix of deep learning and the use of physical laws "only" in the form of a loss/constraint/regularizer -- they often strive for the "explainability" of analytical solutions. (I leave this in the "strengths" section, though it is obviously a talking point that could also go into the weaknesses, and which I suggest to tackle in a potential revision.)
The selection of methods and description of their combination is again well presented; it is too short to introduce into the complex field, but referenced well enough to guide the interested reader to relevant publications. The GNN is mapped to all nodes in the mesh after an embedding layer.

**Weaknesses:**

Bits and pieces I found unclear:
* The source map for the synthetic data is apparently completely random both in shape and in source pressure -- correct? If so, it might be stated. I was puzzled here, since I thought that this is more systematic in nature. I haven't worked in the field of cardiology, though, and can't judge this.
* The source maps were also generated randomly for the real data, so that only the shape of the myocardium seems to be different. Wasn't it possible to obtain data with a real segmentation of the coronary arteries and a better, more realistic estimate of their support regions?
* Modeling-wise: the vascular tree is not a mesh; so treating the entire myocardium like a sponge with arbitrary perfusion in all directions is perhaps not accurate; possibly it is good enough since the objective is just to simulate the last few millimeters from the ends of the vessels into the muscle fibers? My picture of the process, though, suggests that if a region with less supply is downstream of one with more supply, it will get more source pressure as the perfusion in the over-supplied region can't take up the supply -- the GNN on the other hand has only short-distance (next neighbor) interaction. Again, my mental picture might be wrong. I would like to hear a few words more on how the simplistic modeling approach of random source maps can be motivated clinically, though.
* It wasn't clear to me why the very few node features needed to go into the node encoder.

**Detailed Comments:**

All covered above.

**Justification Of The Preliminary Rating:**

The paper is well written, clear, and shows a combination of methods that are in my opinion a door opener for a variety of relevant simulation tasks in the medical field. Be it biophysical modeling of soft tissue deformation -- notoriously hard to parameterise properly and time consuming to actually calculate, or blood flow, or as here, perfusion: pointing out the need for research as this paper does for me is worth to have it presented to the community, hoping that it sparks some interest and discussion.

Weak accept since I'm too little of an expert in this particular combination of methods, and in the underlying medical field to appreciate the contribution fully.

**Questions To Address In The Rebuttal:**

As I advocate acceptance of the paper as an oral, I don't have strong needs for a rebuttal.

**Special Issue:**

No

---

> ### Author Response · Authors · 2024-03-18
>
> We thank the reviewer for her/his insightful comments and positive feedback. We found a mistake in our code after the submission. Fixing the problem significantly improved the results. The modification was added to the revised version of the paper. All the comments are addressed below in the same order as they appear in the review:
>
> - Indeed, the p_source maps and geometries are generated by random processes. We have clarified this point in the revised version of the paper.
>
> - We agree that the model should be tested on more realistic p_source maps. Further work is however needed to collect realistic p_source maps, e.g.  from the coronary artery segmentations and computational fluid dynamics simulations. We believe this issue is beyond the scope of the present paper.
>
> - Indeed, the single compartment Darcy model should be accurate enough, as it aims at simulating only the blood flow in the small vessels such as arterioles and capillaries. A much more detailed vascular tree can be simulated from the end points of the segmented vessels (Jaquet et al.). Nonetheless, this perfusion model delivers accurate results, if it is coupled with blood flow simulation in the larger coronary arteries, using Navier-Stokes equations or reduced models of blood flow, as described in section 2.1. The coupled model requires iteratively solving flow in the vascular tree and in the myocardium, with different p_source maps obtained at each  iteration, until convergence, so that blood flow is well distributed between far-away regions. We refer the reviewer to the reference Papamanolis et al. for more information about the coupled model. Even though classical GNN only
>  have short-distance interaction, the Graph Attention U-net-like architecture we use allows for more distant interactions between nodes.
>
> - The main advantage of the random p_source map is that they are quite easy to generate, and requires no additional simulation or segmentation, compared to more realistic ones. Those synthetic p_source maps were however generated within physiological ranges of p_source from Papamanolis et al. Their spatial heterogeneity is higher than the one derived from the coupled model described in the previous answer. Therefore, we believe that, if our model can predict accurate solutions on these synthetic maps, it should be able to generalize well on realistic ones.
>
> - We agree with the reviewer that, with regard to the small number of node features, the node encoder could appear unnecessary. However, experiments do show that results are better with it. One possible explanation is that it could encode the relative position of the nodes to some specific region of the mesh ( this is plausible as the different meshes have overall similar shapes characteristics, and that b_source is proportional to the mesh volume).  We believe that it can help a node take into account whether or not it is close to a boundary, or to the apex, as the finite element, and therefore, the FV control volumes can be different.

---

### Official Review · Reviewer_tZAV · 2024-02-28

**Confidence:** 4
**Preliminary Rating:** 2
**Recommendation:** Poster
**Final Rating:** 3.5

**Summary:**

This paper presents a finite-volume based GNN to learn PINNs for Darcy equations to simulate myocardial perfusion.  The  PINN leverages the discretized formulation of  the Darcey equation over the tetreahedron mesh of the myocardium, as well as a GNN representing this geometry. The model were trained on synthetic datasets comprising simple half-ellipsoidal shells, and tested on myocardial meshes segmented from CCTA mage from patients.

**Strengths:**

The proposed work represents a step forward for 3D PINNs on complex geometry that have been limited in literature.

The training and test datasets overall considered variability in the geometry including those reflecting actual human subjects.

**Weaknesses:**

The paper is missing some critical details about the methodology that prevents a proper assessment of the work proposed. See detailed comments below.

Another primary concern of the paper is the lack of proper PINN baselines to assess the contribution for the work proposed.

Numbers on computational cost for FVM vs. FVINN are necessary.

**Detailed Comments:**

An important confusion that could be clarified is the discussion about the training and validation sets, as PINNs are typically trained to solve a PDE for a given set of parameters and initial/boundary conditions. How do we train and test on different simulation cases is not clear. Pertaining to this, the coefficients a and b in equation (8) are derived from (7) and should contain information about the patient-specific parameters beta’s as well as those about the p_source and p_sink. That means the training loss are specific to these values that could change from case to case / patient to patient. How are these addressed in the training and validation datasets where these parameters are likely to change among cases?

**Justification Of Final Rating:**

The authors clarified a critical question for me from the original submission, regarding how the PINN can be trained for different geometry and source at the same time. This strengthens the novelty of the work. I'm thus increasing my rating.

**Justification Of The Preliminary Rating:**

This paper represents progress towards enabling PINN on 3D geometry mesh, although some critical details are missing that prevents a good understanding of the methodology as well as the experimental results presented. These need to be addressed for a proper assessment of the contribution of the paper.

**Questions To Address In The Rebuttal:**

Please address the above concern about the methodology of the work (loss in (8)) and the training-testing setup.

Please add proper PINN baselines to demonstrate the benefit of 1) FV-informed formulations and 2) GNN formulations. If it is not possible to do a naive PINN on this, elaborate on why.

Please add computational cost for a FVM simulation vs. FVINN training to obtain the same 3D results.

**Special Issue:**

No

---

> ### Author Response · Authors · 2024-03-18
>
> We thank the reviewer for their insightful comments and positive feedback. We would like to inform the reviewer that a mistake was found in our code after the submission. Fixing the problem significantly improves the results. The changes was added and discussed in the revised version of the paper. We reply to the comments below in the same order as they appear in the review:
>
> - Our FVINN can be trained on varying geometries and p_source maps both at once. This is not possible with classical PINNs. The trained FVINN model is then able to make predictions on new geometry with new boundary conditions at inference time. Indeed, similarly to supervised learning where the loss is computed using multiple ground truths as examples, the loss in (8) comprises patient-specific elements such as p_source values. The model learns a set of convolutional operations which can minimize the FV-informed loss starting from multiple p_source maps and geometries. The main difference between our method and classical supervised methods is that we use a more constrained loss function from FVM. Using these additional constraints, our model can predict a solution without any ground truth supervision.
>
> - Similarly to the previous answer, we believe that our FVINN can not be directly compared to a PINN baseline. However, we added in the revised paper a comparison between FVINN and a classical supervised model. FVINN achieves comparable results with the supervised model, while not requiring any ground truth obtained from classical FEM/FVM solvers or actual measurements. Moreover, the results on the test dataset seem to indicate that FVINN has more robust generalization capabilities compared to the supervised method. The GNN formulation is implemented in this paper in order to predict Darcy solutions on unstructured grids like a FVM and infer on different geometries. To the best of our knowledge, GNN is the only common architecture which can be applied to unstructured grids. We refer the reviewer to the introduction of the paper for the explanations and resources on the advantages of GNN for our application.
>
> - Indeed, our article lacked a computational cost comparison between the two methods. The computational time to solve the linear system of equations given in (7) with the GPU-enabled pytorch function linalg.solve is approximately 4.28 seconds. The forward pass of our network to compute the solution only takes 0.56 second. The two experiments were conducted on the same GPU. This comparison was added to the revised version of the paper.

---

> > ### Comment · Reviewer_tZAV · 2024-03-19
> > **Follow-up questions**
> >
> > Thanks for your clarifications. Please elaborate how the proposed FVINN can be trained on varying geometries and p_source maps at once? In particular, how is the model trained as parameters a and b in the loss function (8) changes among samples, and how do new geometry or parameters a/b get incorporated to get a new PINN solutions on new geometry and p_source? These are critical innovations of the proposed method and are not clear from paper descriptions.

---

> > > ### Author Response · Authors · 2024-03-26
> > >
> > > FVINN can be trained on varying geometries and p_source maps at once as the node coordinates and p_source values are the input of our network. The goal of the proposed method is to find a set of convolutional operations that can reduce the loss function (8) for every node, knowing the position of node neighbors and their p_source values. Once we have learned the set of convolutional operations on the training set, we can apply it to new unseen geometries and p_source maps to predict a solution. The training and inference strategies are similar to classical supervised methods. In a way, given the mesh as input, the model needs to learn the coefficients a/b (which are computed using only finite element geometrical features and p_source values) so that it can successfully minimize the loss function during training and predict a solution at inference.

---

> > > > ### Comment · Reviewer_tZAV · 2024-03-26
> > > >
> > > > Thanks for the response. This makes more sense to me now. I'll update my rating.

---

### Meta-Review · Area_Chair_spRA · 2024-04-02

**Recommendation:** Accept (Poster)
**Confidence:** 4

**Metareview:**

This work innovates in designing a physics-guided GNN for 3D myocardial perfusion prediction. Thanks to the physics-constrained loss function, the method requires no ground truth during the learning process. This allows the model to be trained on a wide variety of p_source maps and geometries and therefore encourages its generalization to new cases. Result shows robustness of the model on unseen geometries and thus tends to validate the generalization hypothesis. The authors have successfully answered the reviewers' questions, which clearly improves the quality of their paper.

For all these reasons, I have decided to accept this article.

---

### Decision · Program_Chairs · 2024-04-06

Accept (Oral)